# Early Forecasting Hydrological and Agricultural Droughts in the Bouregreg Basin Using a Machine Learning Approach

Ayoub Nafii [1,2,*], Abdeslam Taleb [1], Mourad El Mesbahi [1,2], Mohamed Abdellah Ezzaouini [2] and Ali El Bilali [1,2]

1    Hassan II University, Casablanca 20000, Morocco
2    River Basin Agency of Bouregreg and Chaouia, Benslimane 13000, Morocco
*    Correspondence: ayoubnafii@gmail.com

**Abstract:** Water supply for drinking and agricultural purposes in semi-arid regions is confronted with severe drought risks, which impact socioeconomic development. However, early forecasting of drought indices is crucial in water resource management to implement mitigation measures against its consequences. In this study, we attempt to develop an integrated approach to forecast the agricultural and hydrological drought in a semi-arid zone to ensure sustainable agropastoral activities at the watershed scale and drinking water supply at the reservoir scale. To that end, we used machine learning algorithms to forecast the annual SPEI and we embedded it into the hydrological drought by implementing a correlation between the reservoir's annual inflow and the annual SPEI. The results showed that starting from December we can forecast the annual SPEI and so the annual reservoir inflow with an NSE ranges from 0.62 to 0.99 during the validation process. The proposed approach allows the decision makers not only to manage agricultural drought in order to ensure pastoral activities "sustainability at watershed scale" but also to manage hydrological drought at a reservoir scale.

**Keywords:** Bouregreg basin; artificial neural network; hydrological drought; agricultural drought; SPEI





## 1. Introduction

Water resource planning and management in semi-arid regions are confronting several constraints, such as frequent drought phenomena. This natural hazard can impact several ecosystems, such as climatic, hydrological, agricultural, and ecological systems, that can impact socioeconomic activities [1,2]. Indeed, drought is generally caused by hydro-climatic anomalies that result in water shortage in some (or all) processes involved in the hydrological cycle. Various climate variables and their combination can influence droughts, such as an increase in evaporation in the atmosphere, and evaporative atmosphere demand, which lead to water stress [3]. Consequently, as defined by [4], drought is a process in which the hydrological cycle reaches its limit that stresses the related ecosystems. In recent years, droughts have impacted many regions of the world. For example, drought events in Syria [5] and Pakistan [6], the Millennium Drought in Australia (2001–2009) [7], and drought events in the USA, namely: Texas and the Central Great P(2012) and California (2012–2015) [8]. Consequently, the evaluation and prediction of this natural hazard are crucial in water resource management.

For decades, various indices have been developed and applied to evaluate drought risk, such as the standardized precipitation index (SPI), which relies on long-term rainfall records to quantify water scarcity during different time scales [9]. In contrast, the Standardized Precipitation Evapotranspiration Index (SPEI) measures the impact of an increase in temperature on drought by including an evapotranspiration component in the calculation [10]. This approach has been broadly applied to assess and monitor drought episodes in several regions of the world [3,11–14]. Besides, the use of spatial techniques is helpful in quantifying and classifying drought severity by location [15–17].

However, early prediction of the drought index is challenging in water resource planning and management processes, as it allows the decision makers to mitigate the impacts and ensure the sustainability of the related socioeconomic activities. Machine learning algorithms are promising methods for modeling complex hydrological phenomena [18–20]. Drought prediction is a field in which machine learning can present good results, as it requires less time and minimal inputs, and it is relatively less complex than physical or dynamic models [21,22]. For instance, Ref. [23] applied Long Short Term memory to predict SPEI in Australia. Thus, the authors found that the developed models were accurate, with a coefficient of correlation of about 0.99 [24]. They compared the Gaussian Process regression with two ML models and showed its potential accuracy to forecast the SPEI in Iran. Similarly, the Heuristic algorithms, namely: hybrid Adaptive Neuro-Fuzzy Inference System (ANFIS) combined with particle swarm optimization (PSO and genetic algorithm (GA), have been investigated and demonstrated to be accurate approaches to predict drought index SPI [25]. Also, Tree-Based ML models were applied to drought risk assessment and monitoring processes using satellite datasets for different climatic regions [26,27]. The authors found that Random Forest models presented high accuracy compared to Boosted Regression and Tree models with $R^2 = 0.93$. Moreover, it has been reported that the artificial Neural Network (ANN) and Deep Neural Network (DNN) are outstanding approaches in ML models in water resource studies [28–30]. In [31], the researchers concluded that the DNN model outperformed the support vector machine and ANN optimized with GA with an accuracy of about 95% in predicting and assessing drought risk. Table 1 presents the most recent and relevant references published in the literature for predicting drought. Interestingly, the application of data driven techniques showed high accuracy for forecasting drought index in several regions in the world and for different climates.

**Table 1.** Recent works of drought prediction using machine learning.

| Reference | Model/Method | Drought Index | Performances | Country |
|:---:|:---:|:---:|:---:|:---:|
| [32] | Bagging, Random Forest, Random Subspace, Random Tree | SPI | Radom Tree outperformed other models | Syria |
| [33] | Spatial and temporal variation of sustainability in response to meteorological droughts | SPI | *** | Afghanistan |
| [24] | Three machine learning, MLP, GRNN, and Gaussian process regression (GPR) | SPEI | GPR outperformed other models | Iran |
| [34] | ANN, SVM, ANFIS, Decision Tree, | SRI | SVM outperformed other models | Algeria |
| [35] | ANN, SVM | SHMI | Both models showed accurate results | Slovakia |

The water department in Morocco relies on water surfaces to supply water for agricultural, drinking, and industrial purposes through several dam reservoirs, as the groundwater is under continuous overexploitation. The Bouregreg is a typical basin in the country, as it supplies drinking water for the main coastal cities, especially Rabat and Casablanca. However, the water availability in this zone is confronted with several constraints, like frequent drought as an impact of climate change [36] and environmental issues [37]. The drought phenomenon in the Bouregreg Basin is threatening both water availability at the reservoir scale for drinking purposes and the sustainability of the socioeconomic activities related to agropastoral activities at the watershed scale. Consequently, forecasting hydrological and agricultural drought risk is valuable to manage water resources in this area. In this study, the ANN algorithm is used to predict the severity of drought at the end of the hydrological year (month of August) through SPEI. Thus, the timescale used is one month (SPEI-1). Eleven models (ANN_Sep to ANN_Jul) starting from the month of September to the month of July were trained and validated using the data of precipitation

and temperature of twelve stations to predict the annual drought (in August). Then, the relation between the SPEI and the water inflow rate into the reservoir was established to predict hydrological drought risk.

## 2. Materials and Methods

### 2.1. Study Area

The study area is the Bouregreg Basin, which is located between Rabat-Salé-Kenitra, Casablanca-Settat, and Beni-Mellal-Khénifra provinces, Morocco. This basin covers 9975 km$^2$ and consists of 4 watersheds; 4 main rivers compose the hydrographic network and they are named Bouregreg River (264 km), Grou River (249 km), Korifla River (139 km), and Mechraa River (132 km). Climatically, the area is a Mediterranean semi-arid region with an annual average of precipitation of about 400 mm in its North West part and 760 mm in the mountainous part. As for the temperature, it varies between 35 °C and 45 °C during the summer and ranges from 5 °C to 15 °C during winter periods [38]. The reservoir of the SMBA dam, which controls this basin, presents a normal capacity of 975 hm$^3$; it supplies drinking water to urban and rural areas in the coastal area between Sale and Casablanca cities, where the mean inflow rate is about 680 hm$^3 \cdot$r$^{-1}$. Meanwhile, at the watershed scale, pastoral and agricultural activities are the main socioeconomic activities related to water availability.

### 2.2. Datasets

#### 2.2.1. Precipitation and Temperature

The studied basin is monitored by nine rainfall monitoring stations, as presented in Figure 1. In this study, monthly datasets related to rainfall recorded from the period from 1971 to 2021 were provided by the River Basin Agency of Bouregreg and Chaouia (AB-HBC). Table 2 presents the statistical characteristics of these data, such as mean, minimum, maximum, and standard deviation (STD). From this Table, it can be observed that high variability of rainfall is noticed in the high standard deviation for all stations, which can lead either to flood events or drought disasters. The temperature data were recorded at six climatological stations (Figure 1) where three of them are managed by the Bouregreg and Chaouia Hydraulic Basin Agency (ABHBC) and the other three are operated by the National General Direction of Meteorology. Table 3 presents the statistical characteristics of the temperature datasets used in this study.

**Table 2.** Statistical characteristics of the monthly precipitation data recorded at nine monitoring rainfall stations.

| Hydrological Station | River | Period | Mean (mm) | Max (mm) | Min (mm) | Standard Deviation |
|---|---|---|---|---|---|---|
| Aguibat Ziar | Bouregreg | 1976–2021 | 269.5 | 725 | 0 | 198.9 |
| Ras Fathia | Grou | 1976–2021 | 245.2 | 780.7 | 0 | 185.2 |
| S.M. Cherif | Mechraa | 1972–2021 | 232.3 | 785.7 | 0 | 175.5 |
| Ain Loudah | Korifla | 1972–2021 | 225.7 | 671 | 0 | 162.8 |
| Lalla Chafia | Bouregreg | 1971–2021 | 225.5 | 689 | 0 | 166.1 |
| Sidi Amar | Tabahart | 1977–2021 | 225.4 | 657.7 | 0 | 180.1 |
| Sidi Jabeur | Grou | 1971–2021 | 195.5 | 592.1 | 0 | 142.9 |
| Tsalat | Guennour | 1976–2021 | 284.8 | 868.7 | 0 | 212.3 |
| Ouljet Haboub | Grou | 1972–2021 | 184.6 | 563.4 | 0 | 133.7 |

**Table 3.** Statistical characteristics of temperature data of the six climatological stations used.

| Climatological Station | Period of Observation | Mean (°C) | Max (°C) | Min (°C) |
|---|---|---|---|---|
| Rabat | 1960–2021 | 17.5 | 31.6 | 3.9 |
| Khémisset | 1971–2021 | 20.2 | 39 | 2 |
| Khouribga | 1972–2021 | 19.7 | 41 | −1 |

**Table 3.** *Cont.*

| Climatological Station | Period of Observation | Mean (°C) | Max (°C) | Min (°C) |
|---|---|---|---|---|
| Ain Loudah | 1971–2021 | 19.6 | 38.3 | 2 |
| Sidi Jabeur | 1971–2021 | 19.7 | 38 | 2 |
| Ouljet Haboub | 1972–2021 | 19.8 | 39 | 1 |

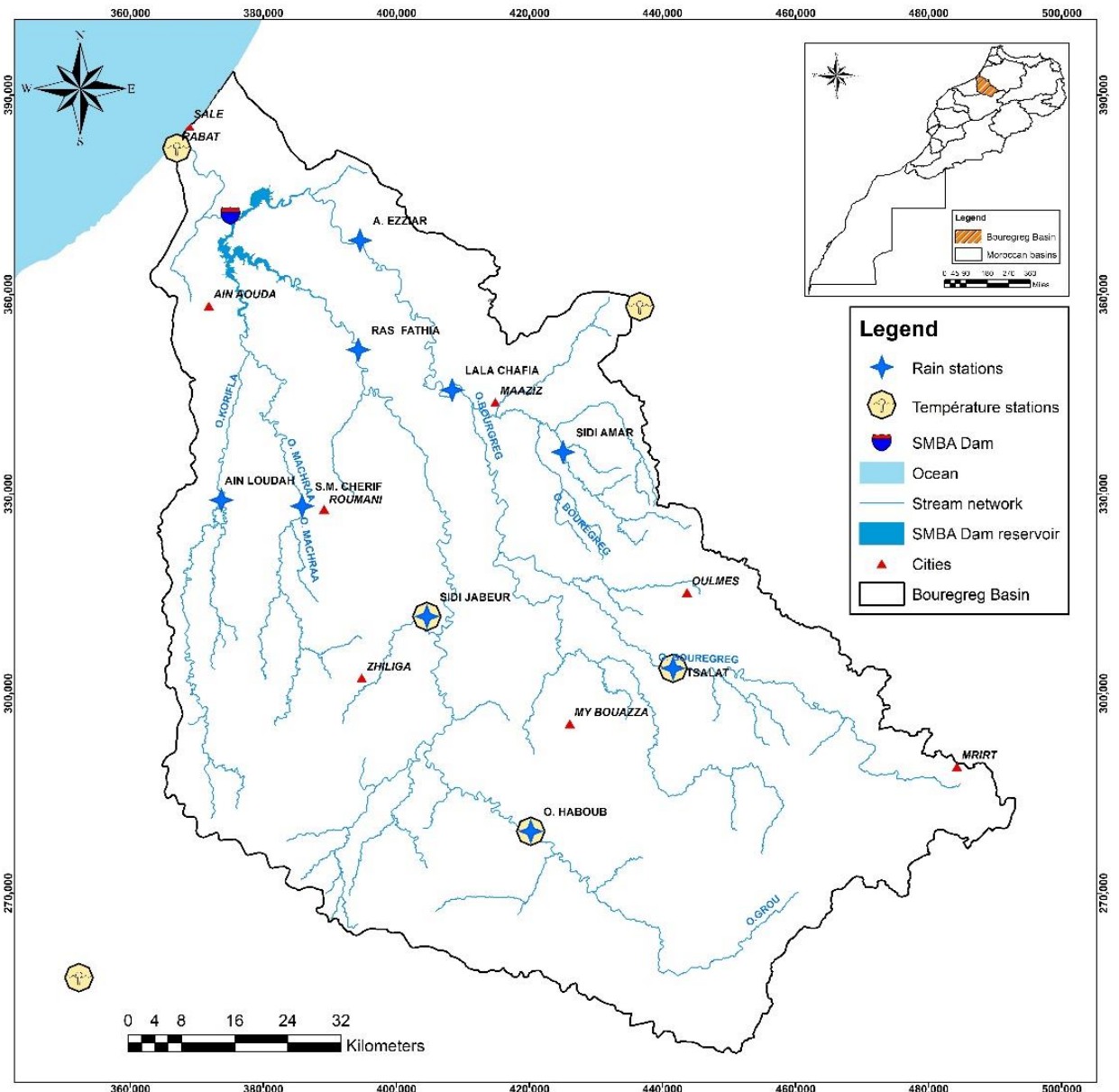

**Figure 1.** Bouregreg Basin location, in Morocco.

### 2.2.2. Runoff Data

The annual inflow data were measured in the SMBA dam for the period from 1985 to 2021 and provided by the Bouregreg and Chaouia Hydraulic Basin Agency (ABHBC). Table 4 presents the statistical characteristics of the whole dataset related to the inflow rate recorded at the SMBA dam reservoir. From this Table, it was observed that high variability in the water inflow rate was noticed in the high value of the standard deviation, which indicates recording drought and flood events.

**Table 4.** Statistical characteristics of the inflow rate dataset of the SMBA dam.

| Inflow Station | Max Annual Inflow (Mm$^3$) | Min Annual Inflow (Mm$^3$) | Mean Annual Inflow (Mm$^3$) | Standard Deviation |
|---|---|---|---|---|
| SMBA's Dam | 2583.9 | 67.2 | 541.9 | 530.6 |

*2.3. Methodology*

2.3.1. Standard Precipitation Evapotranspiration Index (SPEI)

The SPEI is one of the recently developed drought indexes [10]; it shows a growing consensus about its use because it uses rainfall and temperature data rather than only rainfall, as in the case of the Standard Precipitation Index (SPI). It uses the difference between precipitation and evapotranspiration to represent the regional drought. The SPEI uses a simple water balance calculation, and the potential evapotranspiration (PET) is based on the Thornthwaite (1948) model. It has the potential to track agricultural drought more efficiently.

$$Di = Pi - PETi \tag{1}$$

where: $Di$: Deficit in mm, $Pi$: Precipitation in mm, and $PETi$: Potential evapotranspiration in mm for the month $i$.

The calculated $Di$ values are aggregated to a 1-month time scale, Log-logistic distribution is used to model the $D$ series, and the probability density function of a 3-parameter Log-logistic distributed variable $x$ is expressed as:

$$f(x) = \frac{\beta}{\alpha} * \left(\frac{(x-\gamma)}{\alpha}\right)^{\beta-1} \times \left(1 + \left(\frac{(x-\gamma)}{\alpha}\right)^{\beta}\right)^{-2} \tag{2}$$

where, $\alpha$, $\beta$, and $\gamma$ are the parameters of the Log-logistic distribution, and they are obtained using the L-moment procedure (Ahmad et al., 1988), following Singh et al. (1993):

$$\beta = \frac{2\omega 1 - \omega 0}{6\omega 1 - \omega 0 - 6\omega 2} \tag{3}$$

$$\alpha = \frac{(\omega 0 - 2\omega 1) \times \beta}{\Gamma\left(1 + \frac{1}{\beta}\right) * \Gamma\left(1 - \frac{1}{\beta}\right)} \tag{4}$$

$$\gamma = \omega 0 - \alpha \Gamma\left(1 + \frac{1}{\beta}\right)\Gamma\left(1 - \frac{1}{\beta}\right) \tag{5}$$

where $\Gamma$ is the gamma function.

In the Vicente-Serrano et al. [10] study, the probability weighted moments (PWMs) method was used to calculate $\alpha$, $\beta$, and $\gamma$ parameters, based on the plotting-position approach (Hosking, 1990), where the PWMs of order s are calculated as:

$$\omega s = 1/N \sum_{i=0}^{N} (1 - Fi)^S \times Di \tag{6}$$

where: $i$ is the range of observations, $Fi = (i - 0.35)/N$ is a frequency estimator calculated following the approach of Hosking (1990).

The SPEI value is obtained as the standardized value following the classical approximation of Abramowitz and Stegun (1965) [39] given as:

$$SPEI = W - \frac{\left(c0 + c1W + c2W^2\right)}{1 + d1W + d2W^2 + d3W^3} \tag{7}$$

where c0, c1, c2, d1, d2 and d3 are constant: c0 = 2.5155, c1 = 0.8028, c2 = 0.0103, d1 = 1.4328, d2 = 0.1892 and d3 = 0.0013, W is calculated using:

$$W = \sqrt{(-2\ln(Pr))}$$

$$\text{For } Pr \leq 0.5 \tag{8}$$

where Pr is the probability of exceedance of a given D value, the cumulative distribution of D being F(D). When Pr > 0.5, it is replaced with the non-exceedance probability (F(D) = 1 − Pr) and the sign of the calculated SPEI is reversed.

### 2.3.2. Artificial Neural Network (ANN)

In the present study, we applied the ANN model to predict drought. ANN has presented potential accuracy compared to other machine learning algorithms, such as the Support vector machine (SVM), Random Forest (RF), and Linear regression (LR) in several studies as it can map complex and nonlinear systems [40,41].

This subsection provides a short description of this approach. Meanwhile, further details on the ML-based models can be found in [42–47]. ANN models are constructed by three layer types, namely: the input layer, hidden layers (HL), and the output layer [48]. They are interconnected through neurons, which are characterized by weight and bias. The weighted input variables summed with the bias of the layer are transformed from the *j*th layer to the *j* + 1th layer by a transfer function (*f*), and so on, until the output [48]. The training phase is repeated by changing the weights and the biases of the layers until good prediction accuracy (root mean square error) is achieved. To simplify this method, let us take a simple model with one HL. The outputs ($Y_k$) are given by the following equation [49]:

$$Y_k = f_k\left(\sum_{i=1}^{m} W_{jk} \times f_j\left(\sum_{i=1}^{n} X_i W_{ij}\right)\right) + W_0 \tag{9}$$

where *n* is the input variable numbers, *m* is the neurons in the HL, *p* is the neurons of the output layer, and *k* is between 1 and *p*, $W_0$ is the bias, $W_{jk}$ and $W_{ij}$ are the weights between the *j*th neuron and the *k*th output neuron and between the *i*th neuron and *j*th neuron, respectively, whereas $f_k$ and $f_j$ are the transfer functions of the neurons *k* and *j* of the output and hidden layers, respectively. Figure 2 presents the architecture of an ANN model with three hidden layers.

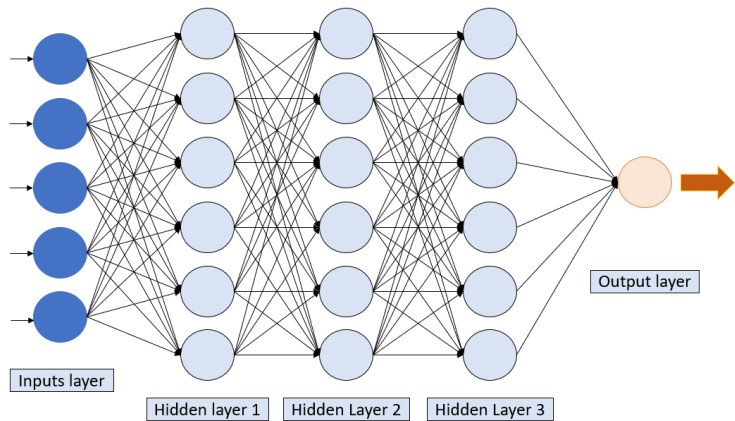

**Figure 2.** Architecture of ANN ML model.

The input data are used to calculate the SPEI at a 1-month time scale in all the stations. The results are used first to characterize the drought in the basin and second to develop models that can predict the annual SPEI so the data is decomposed into two parts: training and validation datasets. Eleven ANN models were developed, trained, and validated; each model predicts the annual SPEI from a corresponding month, from September to

July. Furthermore, the annual SPEI is linked to the annual water inflow rate in the SMBA reservoir to predict and evaluate hydrological drought at the reservoir scale (Figure 3).

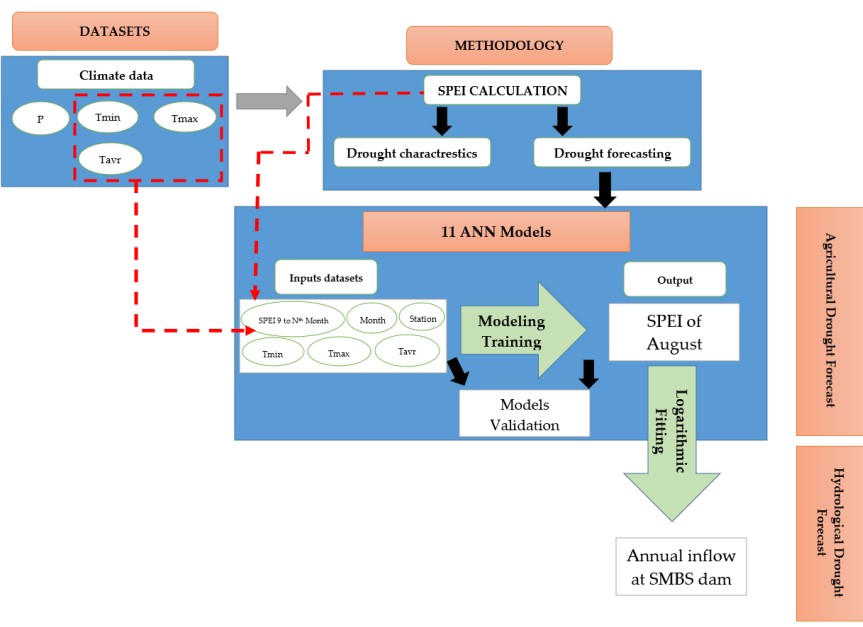

**Figure 3.** Flowchart of the adopted methodology.

2.3.3. Performances of the Models

In this study, we used three statistical metrics to assess the applied ML model performances, namely: the Nash–Sutcliffe model efficiency coefficient (NSE) [50], root mean square error (RMSE), and correlation coefficient (R). Accordingly, an NSE value of 1 indicates a perfect-fit model, greater than 0.75 is a very good fit, between 0.64 and 0.74 is a good fit, between 0.5 and 0.64 is a satisfactory fit, and less than 0.5 is an unsatisfactory fit [51]. However, these performance statistics are defined as:

$$R = \sum_{i=0}^{n} \left( Oi - \bar{O} \right) (Pi - P) / \sqrt{ (\sum_{i=0}^{n} (Oi - \bar{O}i)^2 )) (\sum_{i=0}^{n} (Pi - P)^2 ))} \tag{10}$$

$$NSE = 1 - \frac{\sum (Pi - Oi)^2}{\sum \left( \bar{O} - Oi \right)^2} \tag{11}$$

$$RMSE = \sqrt{\sum_{i=0}^{n} (Pi - Oi)^2 / n} \tag{12}$$

where *Pi* and *Oi* are the predicted and the actual SPEIs, respectively, $\bar{O}$ represents the average values of the actual SPEI index, and *n* is the number of observations.

## 3. Results

### 3.1. Analysis of the Drought Events and Characteristics during 1970–2021

The SPEI was calculated and its normal distribution was fitted and presented in Figure 4. From this Figure, it was observed that 68.3% of the calculated SPEI in all the stations is between −1 and 1, indicating a closer to normal, while 15.9% presented severe to extreme drought. Furthermore, the calculated SPEI shows that several drought events have occurred in the basin, the period of which varies between two and seven years. The seasonal decomposition using an additive model shows clearly a trend and a season in the time series. More importantly, Figure 5 shows the seasonality in the data fluctuation.

However, it is still very weak, causing the SPEI to fluctuate by 0.02 over the course of a year. Besides, the residual shows high variability.

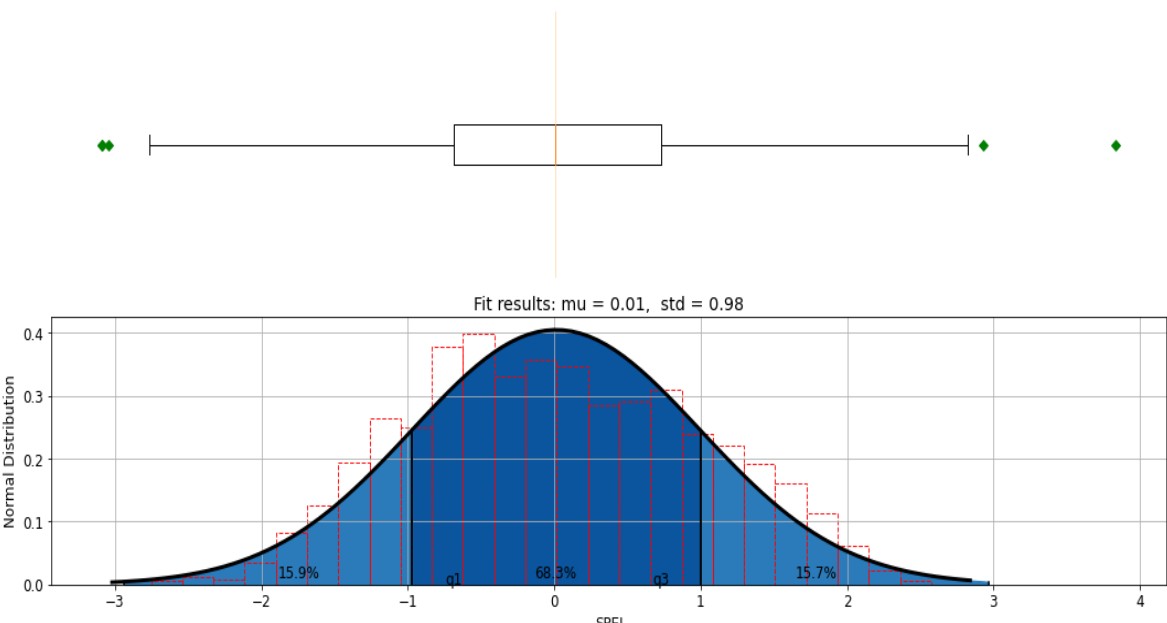

**Figure 4.** Normal distribution of the SPEI data.

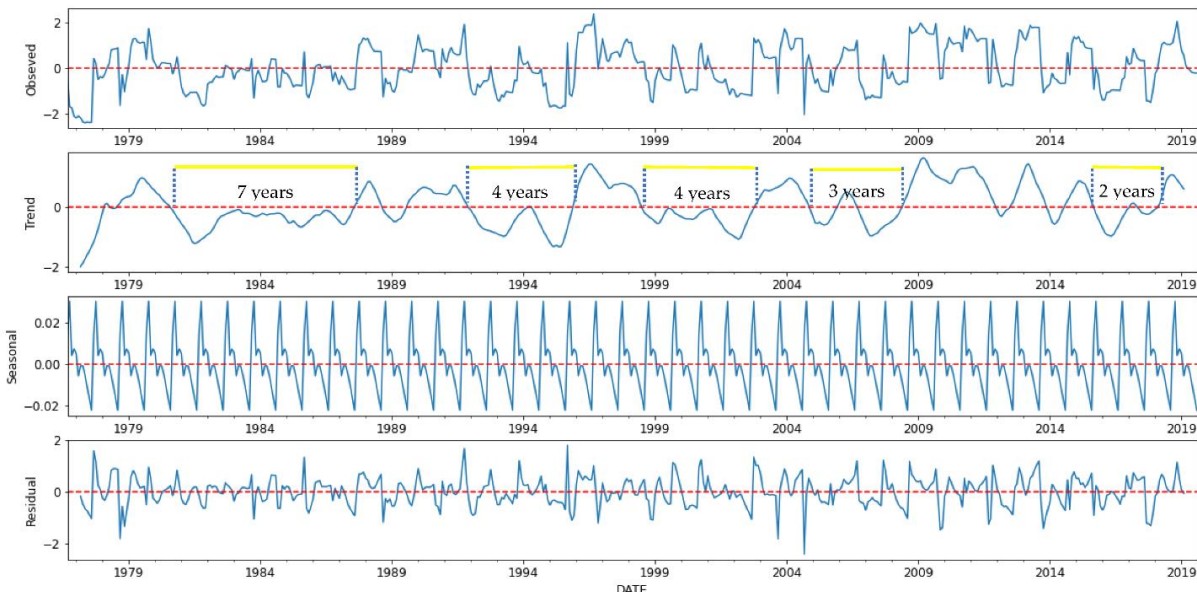

**Figure 5.** Seasonal decompositions applied on the standard Precipitation Evapotranspiration Index (SPEI) at Aguibat Ezziar station.

The Bouregreg Basin has been subjected to critical drought events that have heavily influenced the socioeconomic development in the region by impacting the drinking water supply and agropastoral productivity. As an effect of climate change, the basin can lose an important water yield and, as a result, threaten the sociohydrologic systems [52]. Therefore, early forecasting could be managing drought risks. In this study, we aim to forecast drought using SPEI and ANN.

### 3.2. Data Analysis

The selection of suitable variables is a master key in the development of machine learning models, to guarantee good prediction accuracy and the generalization ability of the models. In this study, the input variables to predict the annual SPEI corresponding to August (of the hydrological and agricultural year) are the corresponding month, SPEI, temperature data: min, max, and mean, and the corresponding station. A correlation analysis of the dataset was carried out in order to explore the potential relationship between the input variables. Figure 6 presents the matrix correlation of the variables for all the stations in the studied basin. It was observed that, except for the mean temperature with the max temperature, all other variables are not highly intercorrelated. Such results prove that the selected variables are not redundant in the prediction of SPEI.

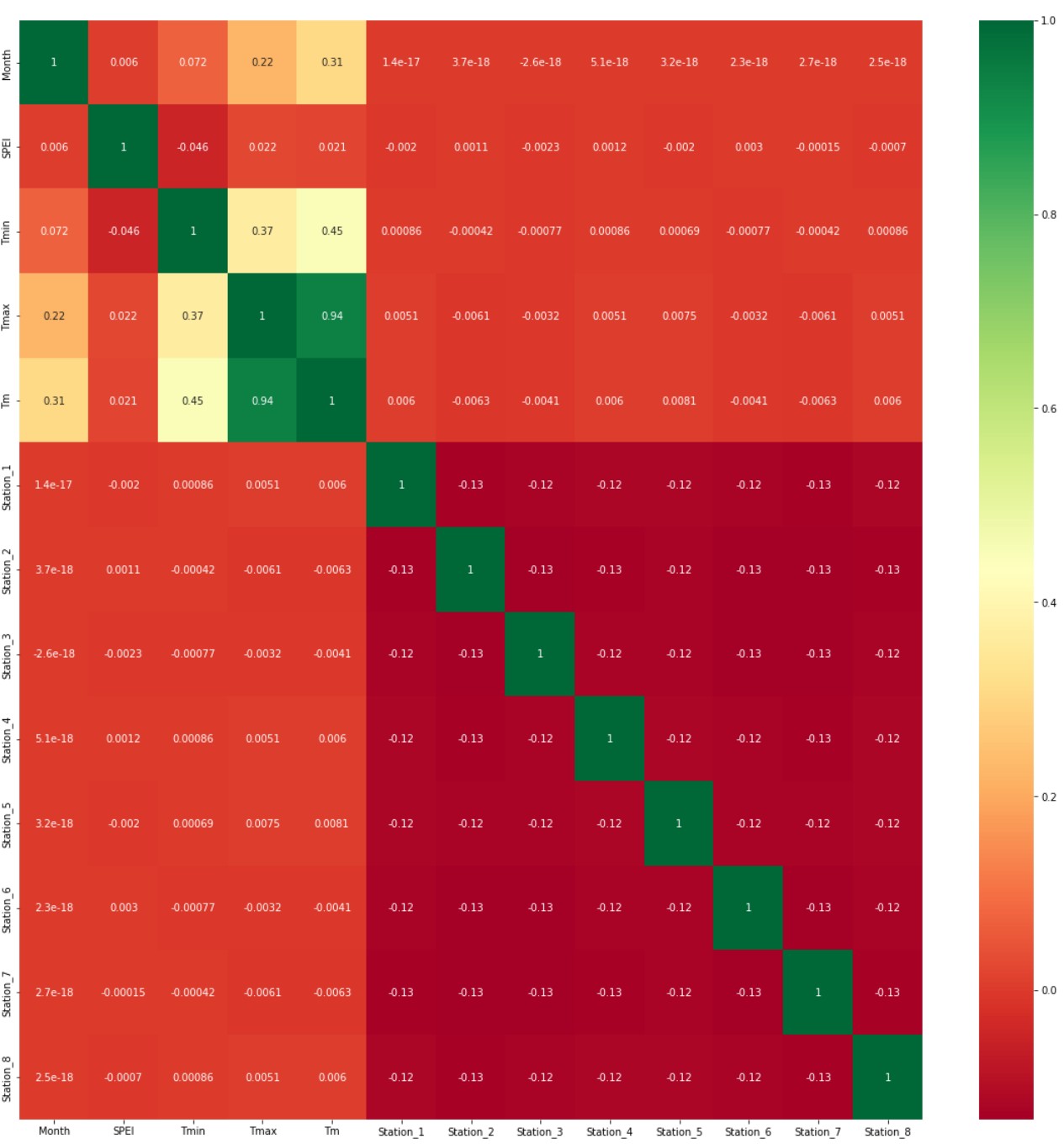

**Figure 6.** Correlation matrix of input data.

### 3.3. Training Process of the Machine Learning Models

In this step, 11 ANN ML models were trained, tuned, and evaluated to improve the prediction accuracy. The tuning process includes changing the hyper-parameters and transfer functions of the models. However, Table 5 presents the adopted parameters in this study.

**Table 5.** Adopted hyper-parameter and functions of the trained models.

| Models | Parameters/Functions/Algorithm |
|---|---|
| ANN_Sep, ANN_Oct, ANN_Nov, ANN_Dec, ANN_Jan, ANN_Feb, ANN_Mar, ANN_Apr, ANN_May, ANN_Jun, ANN_Jul | 6 layers (including input and output layers) 310 neurons in hidden layers (10, 100, 100, 100) Algorithm: Adam Function activation: Softmax Epoch number: 1000 Learning rate: 0.01 |

Table 6 presents the ANN model performances during the training phase. This Table clearly shows that all models have a high potential accuracy in predicting the annual SPEI, justified by the fact that the NSE ranges from 0.64 to 0.99, the $R^2$ ranges from 0.65 to 0.99, and the RMSE ranges from 0.06 to 0.59. It was observed that ANN_Nov to ANN_Jul outperformed the ANN_Sep and ANN_Oct models during the training phase.

**Table 6.** Machine learning model performances during the training phase.

| | $R^2$ | RMSE | NSE | Performance Rank |
|---|---|---|---|---|
| ANN_Sep | 0.65 | 0.59 | 0.64 | Good fit |
| ANN_Oct | 0.73 | 0.51 | 0.73 | Good fit |
| ANN_Nov | 0.82 | 0.41 | 0.82 | Good fit |
| ANN_Dec | 0.81 | 0.43 | 0.81 | Good fit |
| ANN_Jan | 0.87 | 0.35 | 0.81 | Very good fit |
| ANN_Feb | 0.90 | 0.31 | 0.87 | Very good fit |
| ANN_Mar | 0.96 | 0.19 | 0.90 | Very good fit |
| ANN_Apr | 0.96 | 0.18 | 0.96 | Very good fit |
| ANN_May | 0.98 | 0.12 | 0.96 | Very good fit |
| ANN_Jun | 0.99 | 0.08 | 0.98 | Very good fit |
| ANN_Jul | 0.99 | 0.06 | 0.99 | Very good fit |

### 3.4. Validation of the ML Models

This step was carried out to evaluate whether the trained ML models are generalizable in order to predict the annual SPEI for the dataset unseen during the training phase. To that end, we simulated the annual SPEI with 20% of the original dataset and evaluated the model performances. Table 7 presents the models' performances during the validation phase. It was observed that, except for ANN_Sep, ANN_Oct, and ANN_Nov, all other models presented good accuracy during the validation phase, with an NSE ranging from 0.62 to 0.99, an $R^2$ ranging from 0.62 to 0.99, and an RMSE ranging from 0.07 to 0.58. Such accuracy predictions were found by [34] in predicting the standardized runoff index (SRI) in Algeria.

**Table 7.** ANN model performances during the validation phase.

| | $R^2$ | RMSE | NSE | Performance |
|---|---|---|---|---|
| ANN_Sep | 0.35 | 0.77 | 0.32 | Unsatisfactory fit |
| ANN_Oct | 0.36 | 0.81 | 0.25 | Unsatisfactory fit |
| ANN_Nov | 0.37 | 0.81 | 0.25 | Unsatisfactory fit |
| ANN_Dec | 0.62 | 0.58 | 0.62 | Good fit |
| ANN_Jan | 0.75 | 0.46 | 0.75 | Very good fit |

**Table 7.** *Cont.*

|          | $R^2$ | RMSE | NSE  | Performance   |
|----------|-------|------|------|---------------|
| ANN_Feb  | 0.84  | 0.38 | 0.83 | Very good fit |
| ANN_Mar  | 0.90  | 0.3  | 0.89 | Very good fit |
| ANN_Apr  | 0.94  | 0.23 | 0.93 | Very good fit |
| ANN_May  | 0.98  | 0.13 | 0.98 | Very good fit |
| ANN_Jun  | 0.99  | 0.09 | 0.99 | Very good fit |
| ANN_Jul  | 0.99  | 0.07 | 0.99 | Very good fit |

Figure 7 shows that starting from December, we have a linearity distribution in the correlation between the Validation SPEI dataset and Simulation results using the ANN models.

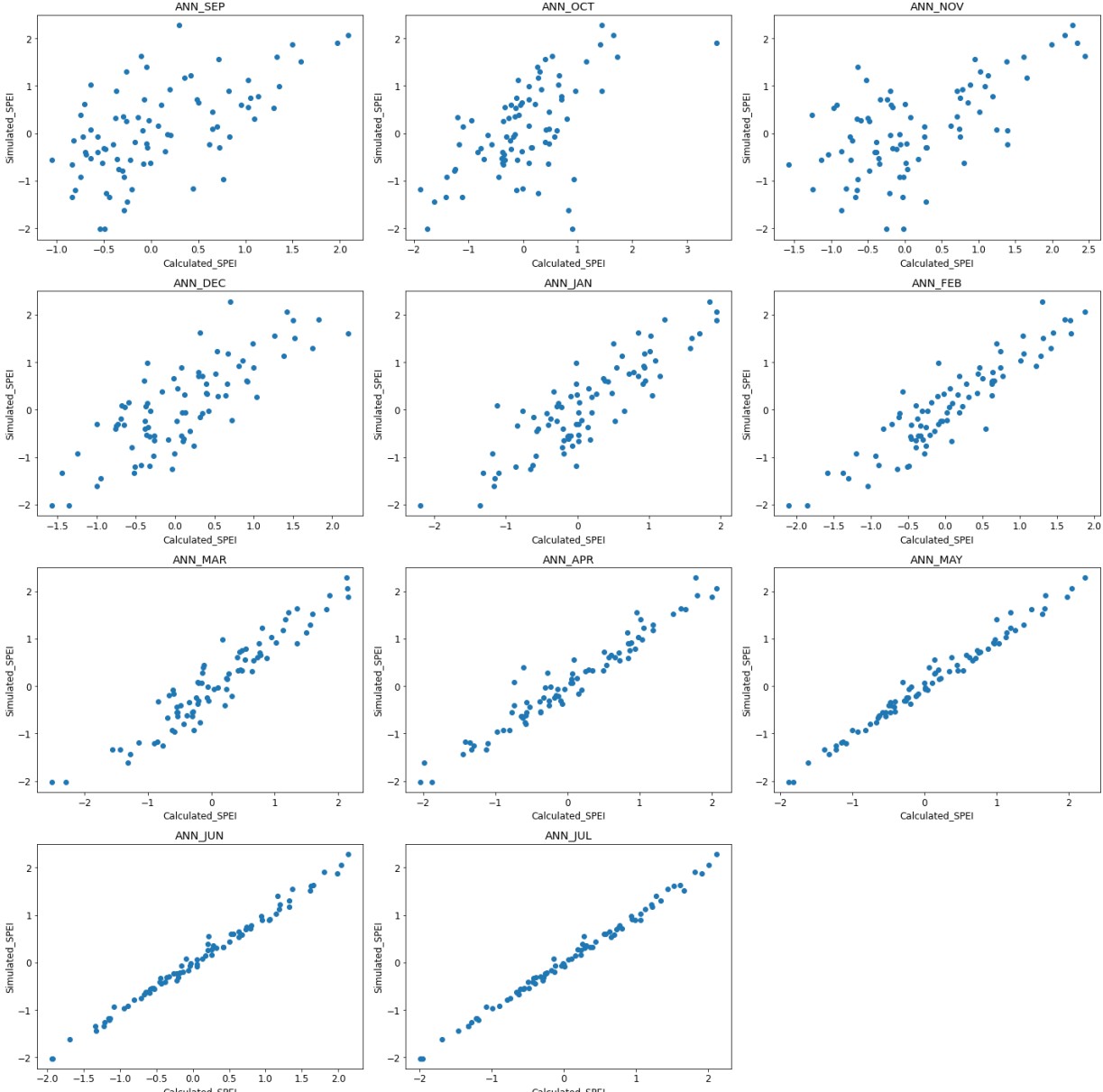

**Figure 7.** Scatter plots of observed vs. predicted.

To determine the models that best predict the annual SPEI, boxplots were developed for the errors (residuals) estimated as the difference between the calculated and simulated values (Figure 8). The presented positive and negative estimation errors indicate underesti-

mation and overestimation, respectively. Globally, from this Figure, the model errors are normally distributed as the median line is closer to zero. However, ANN_Apr, ANN_May, ANN_Jun, and ANN_Jul are the best models for predicting the annual SPEI as it has the lowest error ranges. Thus, the ANN_Mar and ANN_Feb models performed well. Similarly, the models ANN_Dec and ANN_Jan showed fairly acceptable results that can be qualified as acceptable since 75% of the error was between $-0.25$ and $0.25$ and evenly distributed over 0.

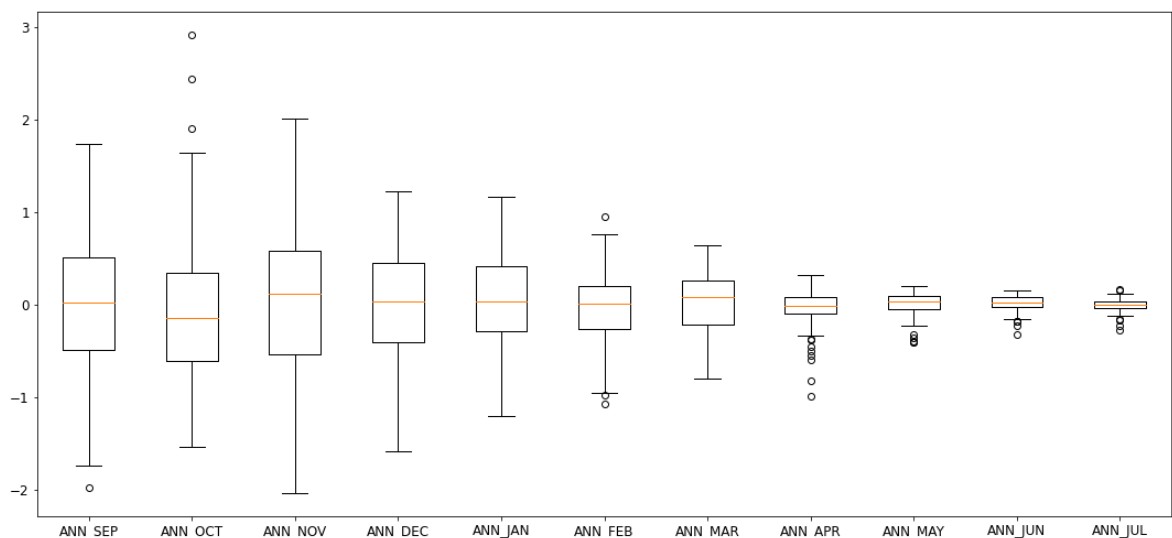

**Figure 8.** Boxplots showing the error distributions of the SPEIs in the test phase. The horizontal line shows the median errors, lower box shows negative errors, circles indicate outlier error values.

Additionally, Figure 9 illustrates the comparison of the observed, and predicted August from September a), from December (b), and from April (c). From this Figure, it was observed that it is not possible to predict SPEI from September (Figure 9a), as there are significant discrepancies between observed and predicted SPEI provided by ANN_Sep model. Meanwhile, from December, the ANN_Dec model is fairly accurate to predict SPEI. Therefore, this model can be useful for predicting agricultural drought early.

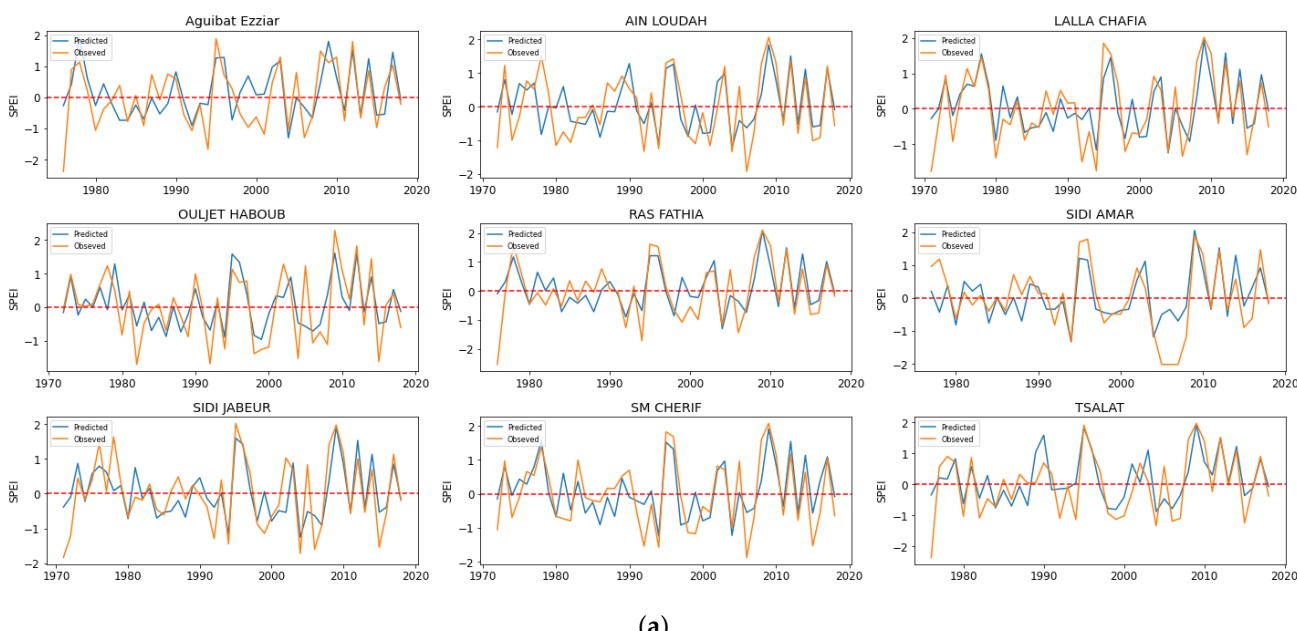

**(a)**

**Figure 9.** *Cont.*

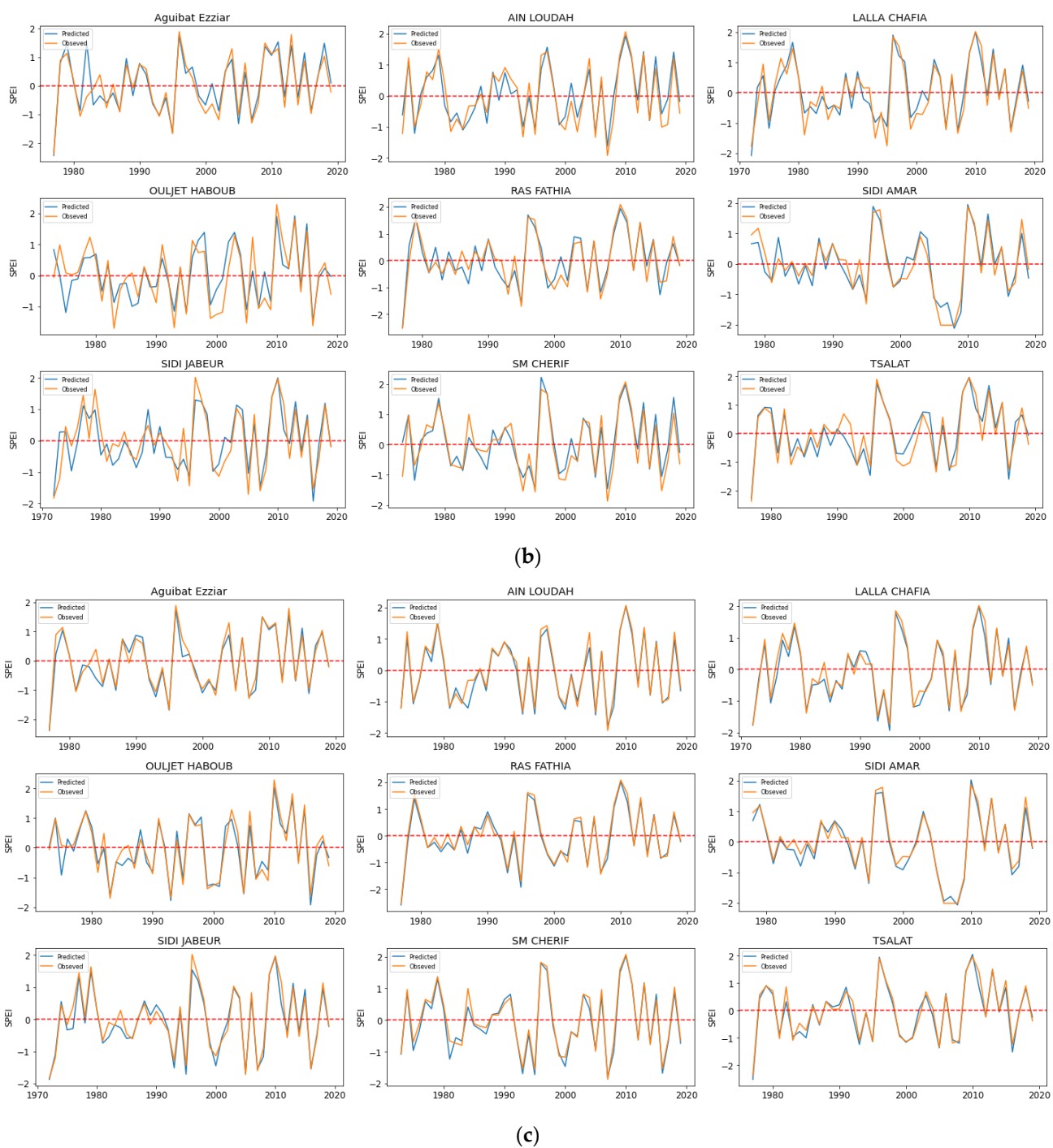

**Figure 9.** Examples of superposed plots comparing calculated and predicted annual SPEI. SPEI prediction of August from September (**a**), from December (**b**), and from April (**c**).

### 3.5. Hydrological Drought Prediction

To manage water drinking supply from the SMBA reservoir, the information on the annual inflow is crucial since the basin controlled by the SMBA dam is characterized by high inflow variability that is dependent on precipitation. Consequently, linking between hydrological and agricultural droughts is valuable to prioritize mitigation measures against drought at watershed and reservoir scales. To predict the annual inflow in the SMBA reservoir, a logarithmic fitting between inflow data and the mean annual SPEI of the nine

stations was carried out using the historical data. The correlation presented very good results with an $R^2$ of 0.88.

Figure 10a shows that the correlation predicts very well the annual inflow using the annual SPEI, indicating that the response of the hydrological drought to the agricultural drought is exponential. Besides, Figure 10b illustrates a comparison between calculated and observed SMBA reservoir inflow values for the period from 1979 to 2021. It obviously shows that there is an agreement between the observed and calculated water inflow rate.

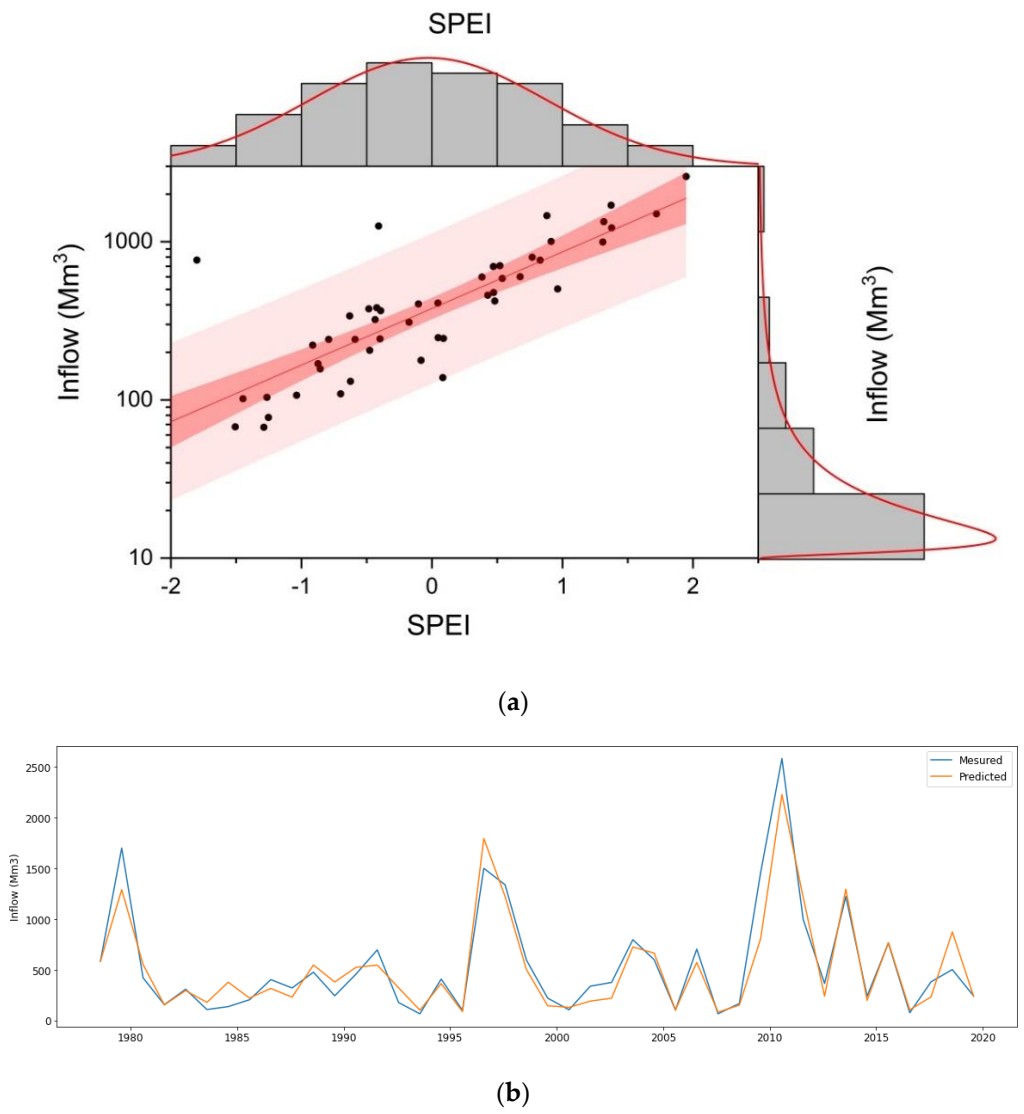

(**a**)

(**b**)

**Figure 10.** (**a**) Logarithmic fitting of inflow date with the monthly SPEI using bivariate distribution, (**b**). Observed and predicted inflow using the logarithmic fitting through semi-logarithm scale.

Drought hazards can lead to a substantial economic loss, especially for countries whose economies rely on the agriculture sector [53,54]. Besides, this natural hazard is a threat to water scarcity for drinking purposes in the semi-arid environment, where most countries rely on reservoirs for water surface mobilization [55]. As for food security, the agricultural drought impacts food security, especially in sub-Saharan countries [56]. Compared to traditional methods based on the evaluation of the instant drought using SPI, SPEI, or NDVI, the early prediction of the annual SPEI is valuable to assess the appropriate measures in time for mitigating the drought consequences and socioeconomic impacts. To reach this overarching goal, ML models represent an innovative approach in hydrological sciences [30,57,58]. In this study, the ANN model provided fairly acceptable

performances for predicting hydrological and agricultural draughts with NSE, about 0.62 from December to 0.99 in July. These results allow the decision maker to predict early the drought impacts (from December) and, therefore, to plan appropriate measures to reduce the socioeconomic impacts. However, using SPEI to evaluate agricultural and hydrological droughts, respectively, is not enough in the situation where agriculture relies on irrigation. In this instance, embedding socioeconomic activities to evaluate drought risk is primordial and, therefore, can be suggested for future works.

Despite being an accurate method to predict hydrological and agricultural annual drought, this approach is unable to predict drought in river networks, which are water source supplies of several ecoservices, such as the environmental flow, water withdrawal for livestock production, and recreational purposes, the combination of the hydrological model with the prediction of SPEI could be valuable to overcome such limitations, and therefore it is suggested for future works.

### 4. Conclusions

Recently, climate change effects combined with socioeconomic activities have led to water scarcity in several regions, particularly semi-arid environments. The drought prediction at the basin scale, like precipitation, is nonlinear and highly dynamic, especially in semi-arid regions. In this study, 11 ANN models with 4 hidden layers, Softmax as a function activation, and a learning rate of 0.01, were applied and evaluated to predict the annual SPEI starting from September to July. The main conclusions of the present study are as follows:

1. Developed ANN models presented good prediction accuracy in forecasting drought using SPEI, with an NSE ranging from 0.62 to 0.99, an $R^2$ ranging from 0.62 to 0.99, and an RMSE ranging from 0.07 to 0.58. Thus, the generalization ability through the validation process demonstrated the stability of the applied models in predicting the annual SPEI;
2. From December, the models are fairly accurate in predicting the annual SPEI at the end of the hydrological year;
3. Hydrological drought is exponentially linked to agricultural drought.

Overall, the study results provide new insight into the early forecast of the agricultural and hydrological drought risks. Implementing this approach in water resource planning and management could be a fruitful tool to manage the drought impacts not only at the watershed scale but also at the reservoir scale for drinking water supply purposes.

**Author Contributions:** Conceptualization, A.N. and A.T.; methodology, A.N.; software, A.N. and M.E.M.; validation, A.N. and A.T.; formal analysis, A.N. and A.E.B.; investigation, A.N.; resources, A.N.; data curation, M.A.E.; writing—original draft preparation, A.N.; writing—review and editing, A.N. and A.E.B.; visualization, A.N.; supervision, A.T. and A.E.B.; project administration, A.T. All authors have read and agreed to the published version of the manuscript.

**Funding:** This research received no external funding.

**Data Availability Statement:** All data used in this study is available on simple request.

**Acknowledgments:** This work is supported by the River Basin Agency of Bouregreg and Chaouia (ABHBC) by providing the required datasets. So, the authors thank the ABHBC teams for their help and assistance during the achievement of this study.

**Conflicts of Interest:** The authors declare no conflict of interest.

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
