# Peer review of "Early Forecasting Hydrological and Agricultural Droughts in the Bouregreg Basin Using a Machine Learning Approach"

_water, doi:10.3390/w15010122_

Round 1

Reviewer 1 Report

Ayoub et al. presented a method on forecasting hydrological and agricultural droughts in Bouregreg basin using machine learning approach. Although the manuscript is well written, proper context are missing in the manuscript. Below are points of attention from this reviewer.

Point 1: The first time a phrase is mentioned that can be abbreviated, spell it out in full and provide the abbreviation in parentheses. Use only the abbreviation thereafter. For example, expand SPI in Line 41 when it’s first introduced in the paper. Similarly, since SPEI has been explained in Line 43, only abbreviation is needed in Line 55, Line 84, etc. And same rule applies to ANN, ABHBC.

Point 2: Authors are advised to explain all parameters (at least mention their names) in equations (1) – (8) in Section 2.3.1. And improve the format of equation (8) (Line 167 - 168).

Point 3: Authors are advised to introduce Figure 2 and 3 in Section 2.3.2 or related section. Putting graphs in the manuscript without introduction is confusing.

Point 4: Improve the subtitle for Figure 4.

Point 5: Table 6 and Figure 7 represent the same idea. Authors are suggested to keep only one of them.

Point 6: Authors are advised to revise the Conclusions section. It needs to be more concise, including the ANN model characteristics and numerical errors.

Point 7: Authors are advised to briefly discuss why ANN is a better choice over other machine learning models in this study. This part can be added to either Section 1 or Section 2.3.2.

Point 8: There are some typo, grammar, and logic issues in the manuscript, such as:

-       Line 14, “machine learning algorithms”.

-       Line 29 -31, two sentences starting with “Indeed”.

-       Line 38, “plains (2012) and California (2012-2015)”.

-       What are two “…” in Line 90?

-       “his” in Line 94.

-       “3” in “hm3” in Line 98 should be a superscript. “hm3.yr-1” should also be corrected.

-       The font size of Figure 1’s, Table 4’s and Table 5’s subtitles are different from other subtitles.

-       Line 112-116 need to be rewritten. For example, “The temperature data were recorded at 6 climatological stations (Figure 1) where 3 of them are managed by the Bouregreg and Chaouia Hydraulic basin Agency (ABHBC) and the other 3 are operated by the National General Direction of Meteorology. Table 2 presents the statistical characteristics of the temperature datasets used in this study”.

-       The “10” in Line 122 should be 9? As there are only 9 hydrological stations in Table 1.

-       “,” in Table 2 and 3 should be replaced by “.”.

-       There is no p (mentioned in Line 185) in the equation in Section 2.3.2.

Authors should read through the manuscript carefully and improve its readability.

Author Response

Thank you for your comments and suggestions. They are very constructive and revealed a good mastering of machine learning related environmental sciences. Indeed, the comments and suggestions significantly improved the revised paper in terms of scientific and academic perspectives. The authors were pleased to take into consideration all comments and suggestions.

Our responses are given in attached file. The addressed issues and suggestions have been taken into consideration in the article (highlighted in the manuscript).

Reviewer 2 Report

The authors attempt to develop an integrated approach to forecast the agricultural and hydrological drought in a semi-arid zone to ensure sustainable agro-pastoral activities at watershed scale and drinking water supply at reservoir scale. This work is very meaningful. However, flood forecasting is very difficult because of its huge uncertainty. Using Machine learning algorithms to forecast SPEI is a useful and common methods.

There is not much innovation in the method of the article, and the degree of analysis is not enough. For example, the flood forecast is carried out by month, and the models are also established by month, but the precipitation degree of each month is not explained, and which months are more significant for the forecast? The manuscript suggest that some months are suitable for forecasting, but is it necessary to forecast these months? In addition, it is important that future floods are not predicted according to the current situation.

In short, I think this work has much promotion space. Some comments are listed as following.

L20. what is “reservoir scale”

L36-38. What are the consequences of the droughts. That is the reason why we need to evaluate and predict the natural hazard are crucial in water resources management.

L46. Please specify what kind of spatial techniques.

L55. The full name of SPEI has been written. Only use abbreviations here. The following part of the manuscript also be like this.

L58. What is ML models? And Tree-Based ML models?

L70. What are the specific contents of data driven techniques? It should be specified in detail.

L77. From my perspective,  “frequent drought” is belongs to “climate change impacts”.

L85. Why use “one month” as the timescale. Is there any basis, data format or experience?

L85. What is 11models? Please introduce 11 models in detail.

L90. “located between …and …”? Some words are missing.

L97 & L100. hm3, hm3.yr-1. Please use superscript number format.

L124 & L126. The font size in the two tables is different.

L138. Superscript number format problem too.

L148. Formula (1). What are the meanings of I, P and PEI? And add the units of D, P and PEI.

L152. Formula (2). What is the meanings of x?

L155. Formula (3). What are the meanings of W0, W1 and W2?

L162. Formula (6). What is the meanings of Ñ¡?, I, N, F, S?

L165. Formula (7). What are the meanings of W, c0, c1, c2, d1, d2 and d3?

L166-167. This P is same as P in formula (1). These two Ps have different meanings. Please change a symbol. And double “For P 0.5”?

L171. “The constants are c0 = 2.5155, c1 = 171 0.8028, c2 = 0.0103, d1 = 1.4328, d2 = 0.1892 and d3 = 0.0013.” Is there reference?

L190. There are three hidden layers in figure 2. Is there three hidden layers in model established by authors? How is the number of HL determined?

L210. Locations of “i=0” and “n” in symbol sigma should be same as other formulas.

L223. What is the meaning of additive model?

L224. Please describe Figure 5 in detail

L226. In figure 5, the residual is showing high variability. This is obviously not a good phenomenon. Based on analysis, how do you prove that your model is feasible?

L232-237. This paragraph is not result analysis.

L241. In this study, the input variables to predict the annual SPEI are the corresponding month, SPEI, Temperature data: min, max and mean, and the corresponding stations. However, the parameters shown in figure 6 do not completely correspond to these, because there is moisture in figure 6, which is not described in the context.

L246. In figure 6, variables, except for temperature, are not highly inter-correlated. “Such results prove that the selected variables are not redundant in the prediction of the SPEI.” This statement is not suitable, because the study did not consider other parameters, and obviously only temperature is the most relevant among these parameters.

L266 & 276. In Table 5 & 6, whether performance effect is good, very good or unsatisfactory depends only on NSE? What are the functions of R2 and RMSE? Is R2 also related to the quality of interval representation effect?

L273. According to the results of Machine Learning Models and ML Models, ANN_Sep, ANN_Oct and ANN_Nov models are not well, what is the reason?

L280. It is recommended that the abscissa scale of all subgraphs be the same as the ordinate, and draw 1:1 lines. That will be more intuitive. In this figure, it can be seen that the prediction effect from Jan to Jul are better than that of other months. What is the reason for this? Is it related to the amount of monthly precipitation? This is a very important issue. The monthly precipitation data is not listed in this paper. I think this should be added and be analysised.

L300. “from December the ANN_Dec model is fairly accurate to predict SPEI”.  End symbol is missing.

L301. Whether predicting agricultural drought early is necessary?

L330. In Figure 10, the Inflow histogram and curve in right part do not seem to match the data points? the fitting line is exponentially type?

Author Response

(The authors gave the same response as above.)

Reviewer 3 Report

- The abstract con include the numerical results obtained.

- A detailed discussion on the novelty and the key contributions of this work can be presented.

- Summarize the recent works in the form of a table.

- Some of the recent state of the art such as the following can be discussed

Capability and Robustness of Novel Hybridized Artificial Intelligence Technique for Sediment Yield Modeling in Godavari River, India.

- How did the authors chose the hyper-parameters in this study? Is it random or any hyper-parameter tuning is applied?

- Compare the results obtained with recent state of the art.

- What is the computational complexity of the proposed approach?

- Discuss about the limitations and future enhancements of this work.

Author Response

(The authors gave the same response as above.)

Round 2

Reviewer 1 Report

All comments from the first round have been address properly. This reviewer suggests acceptance in the present form.

Reviewer 2 Report

It's fine.

Reviewer 3 Report

The authors have addressed the comments well. I have no further queries/suggestions.